# A Systematic Review of Emerging Ventilated Acoustic Metamaterials for Noise Control

Linus Yinn Leng Ang [1,*], Fangsen Cui [1], Kian-Meng Lim [2] and Heow Pueh Lee [2,*]

1. Institute of High Performance Computing (IHPC), Agency for Science, Technology and Research (A*STAR), 1 Fusionopolis Way, #16-16 Connexis, Singapore 138632, Singapore
2. Department of Mechanical Engineering, National University of Singapore (NUS), 9 Engineering Drive 1, #07-08 Block EA, Singapore 117575, Singapore
* Correspondence: linus_ang@ihpc.a-star.edu.sg (L.Y.L.A.); mpeleehp@nus.edu.sg (H.P.L.)

**Abstract:** As sustainability is one of the main pillars in developing future cities, adopting natural ventilation whenever possible is one way to reduce energy consumption, thus indirectly reducing carbon dioxide emissions. Lately, ventilated acoustic metamaterials have started to receive more research attention because of how they can provide both ventilation and noise control. Motivated by this research attention, we present this timely systematic review of emerging ventilated acoustic metamaterials for noise control. By limiting the review to a five-year coverage (2018–2023), three kinds of ventilated acoustic metamaterials were identified—metamufflers, metapanels, and metacages. Based on the studies included in this review, we discuss the present challenges of metacages. More research efforts are still needed to see real-world applications of metacages as a novel ventilated noise control measure in the future.

**Keywords:** ventilated acoustic metamaterial; metamuffler; metasilencer; metabarrier; metapanel; metasurface; metacage; noise control; sustainability

## 1. Introduction

By now, countries around the world will have experienced the effects of climate change. Climate change is caused by the excessive amount of greenhouse gases in the atmosphere. Naturally, the atmosphere contains an adequate amount of greenhouse gases that trap heat to keep the Earth warm. Due to anthropogenic activities, additional greenhouse gases are emitted into the atmosphere. Consequently, the atmosphere traps more heat and warms the Earth further, leading to global warming. Droughts, storms, and heatwaves happen more frequently and are more severe, in addition to the rising sea level.

In the latest report published by the Intergovernmental Panel on Climate Change (IPCC) [1], carbon dioxide emissions have been rising year by year and continue to be the highest among other greenhouse gases. Recent statistics show that as of 2019, 75% of the emitted greenhouse gases in the world are carbon dioxide. Methane ranks second at 18%. Among the various anthropogenic activities, the burning of fossil fuels for energy contributes the most (85%) to the emissions of carbon dioxide. As energy is one of the basic human needs, it is a global challenge to cut down on the burning of fossil fuels as world population increases. As fossil fuels will eventually be depleted [2–4] and decarbonization has received more emphasis, research interest in finding alternatives has recently been growing [5–11].

Due to global warming, it is common for living spaces in tropical and subtropical countries to rely on air conditioners for indoor thermal comfort [12]. As windows are typically closed during the use of air conditioners, occupants of the living space can also enjoy better indoor acoustic comfort. With good indoor thermal and acoustic comfort, the living space can be conducive for work, play, and rest. However, the reliance on air conditioners leads to high energy consumption, contributing indirectly to carbon dioxide emissions.

As sustainability is one of the main pillars in developing future cities, adopting natural ventilation whenever possible is one way to reduce energy consumption in buildings. For example, Weerasuriya et al. [13] reported that natural ventilation could reduce energy consumption in a 40-story-high residential building by up to 25%. In another example, Cardinale et al. [14] made use of simulations to show how natural ventilation could reduce energy consumption in a two-story semi-detached house by up to 18%, depending on the orientation.

Although natural ventilation can still provide good indoor thermal comfort, indoor acoustic comfort is compromised. Outdoor noise can propagate into a living space directly through open windows without much energy loss. According to the literature [15], it is possible to address this problem with plenum windows. Plenum windows are essentially double-glazed windows with staggered openings—one connected to the environment and the other connected to the living space. The staggered openings help to direct outdoor sound propagation along a duct-like path before reaching the living space. Additional structures—such as sonic crystals [16] and labyrinthine arrays [17]—can also be considered along the duct-like path to achieve further noise reduction for specific applications.

Lately, ventilated acoustic metamaterials have started to receive more research attention [18,19]. As the name suggests, this type of acoustic metamaterial may be another potential solution for addressing the same problem. As ventilated acoustic metamaterials only received more research attention over the last five years (2018–2023), it is time to systematically review the state of the art and identify the emerging design of ventilated acoustic metamaterials. Section 2 presents the approach used to conduct the systematic review. Subsequently, the studies included in this review are categorized into three design types—metamufflers, metapanels, and metacages. These three design types are separately discussed from Section 3 to Section 5. Finally, an outlook on metacages—identified as the emerging ventilated acoustic metamaterial—is provided in Section 6 before the conclusion in Section 7.

## 2. Review Approach

This review was performed according to the Preferred Reporting Items for Systematic Reviews and Meta-Analyses (PRISMA) guidelines [20]. The overall process is shown in Figure 1. The records were first identified from scholarly databases and through citation chasing. Secondly, the records were manually screened to ensure relevancy to the topic. Lastly, the full text of each record was reviewed. These three steps are discussed in detail in Section 2.1 to Section 2.3.

### 2.1. Identifying Records

The literature search was performed in two scholarly databases—Web of Science and ScienceDirect. In total, 125 records were identified. By processing the search results in a spreadsheet, eight duplicated records were identified and removed. Google Scholar was not considered because it is currently regarded as an unsuitable professional tool for searching for scientific literature [21]. Citation chasing was also performed to ensure the completeness of this review. This process helped to identify nine more records.

The search strategy included terms related to different forms of the word "ventilate" *(ventilat\*)* , terms starting with "meta" *(meta\*)*, terms related to the topic "acoustics" *(acoustic\* OR noise OR sound)*, and terms related to the purpose *(reduc\* OR control OR mitigat\*)*. The asterisk symbol represents any number of wildcard characters that can form terms starting with the given characters. The literature search was last performed on 20 January 2023.

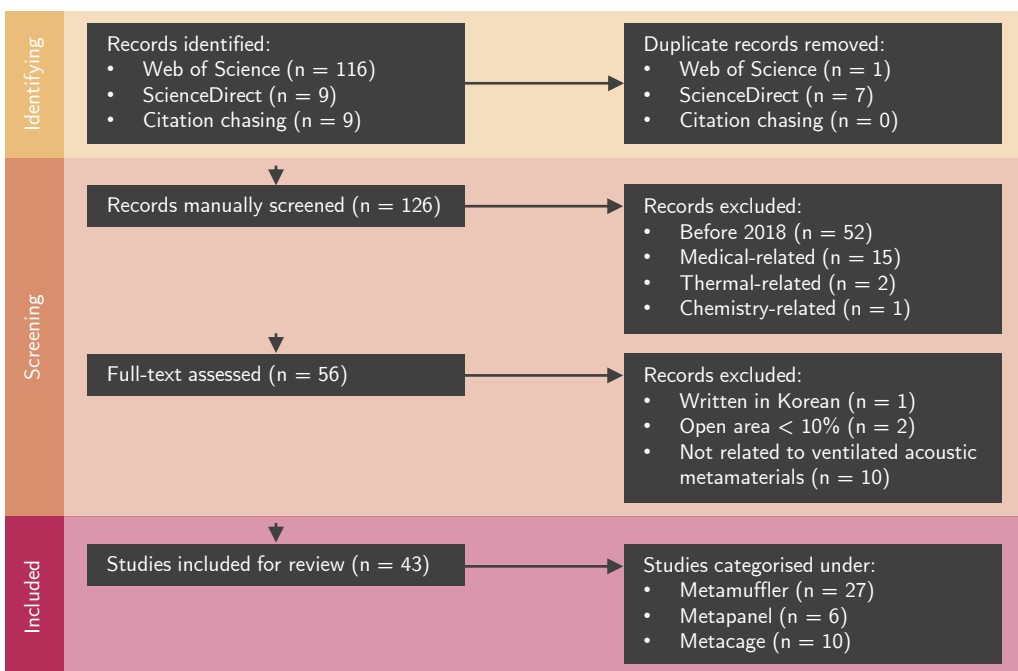

**Figure 1.** Flow diagram showing the approach to performing the systematic review.

*2.2. Screening Records*

To ensure the recency of this review, we considered only peer-reviewed articles published between 2018 and 2023. As a result, 52 records were removed. The remaining 74 records were manually screened by the title, keywords, and abstract. From this screening, 18 records were found to be irrelevant. A full-text assessment was then conducted on the relevant records (56 of them). One record was removed because its full text was written in Korean. Two records were removed because the proposed designs had a small open area of less than 10%. This percentage is believed to provide inadequate ventilation. Finally, 10 records were removed because ventilation was only discussed in the main text, and was not a feature of the proposed design. At the end of this whole process, 43 records remained for inclusion in this review.

*2.3. Included Records*

Following the review of the 43 records, the proposed designs could be categorized into three design types—metamufflers (27 records), metapanels (6 records), and metacages (10 records). Each design type was reviewed in three separate sections. By grouping the 43 records based on the corresponding author's country of origin, we found that research interest in ventilated acoustic metamaterials was mainly in Asia (36 records), followed by North America (4 records), and then Europe (3 records). If the records were instead grouped based on the first author's country of origin, one of the records in Europe would be shifted to Asia. This information can help researchers identify where to establish potential collaborations.

**3. Metamufflers**

For noisy environments in which sound waves propagate in only one direction (i.e., one-dimensional), ventilated acoustic metamaterials were already studied [22–24] long before the five-year coverage of this review. The main reason lies in the convenience of fabricating specimens with rapid prototyping and characterizing them in the impedance tube—an acoustic test device that measures the performance of a given material subjected to plane waves at normal incidence. An impedance tube is not the only option. A waveguide developed in-house can also be considered to characterize specimens that do not fit into the standard diameter (3 or 10 cm) of the impedance tube. As they are small-scale, the material

cost incurred in such studies is generally low. Moreover, the process of characterizing small-scale specimens serves as the foundational work for deciding whether further studies should invest in a large-scale setup. Of course, the intended application may affect the final setup when scaled up. In this case, the small-scale specimens—originally studied as metamufflers—may then be arrayed in certain ways to form a metapanel or a metacage. In Sections 4 and 5, we will better understand why according to the reviewed studies.

As highlighted above, although ventilated acoustic metamaterials for one-dimensional noise control have been studied for more than a decade, the terms "ventilated" and "metamuffler" came about only recently when natural ventilation was identified as one of the many ways to reduce energy consumption in buildings and to promote heat dissipation in machines. In the literature, the term "metasilencer" [25] is sometimes used in place of the term "metamuffler" [26]. Note that both terms refer to the same type of ventilated acoustic metamaterial. The only difference lies in the choice of language (i.e., British or American English). For consistency in writing, the term "metamuffler" is used in this review.

Like traditional duct silencers, metamufflers are implemented by replacing a finite section of the ducting system to reduce flow noise. The noise-reducing mechanism is realized via specially designed features on the interior walls of the metamuffler. Based on the studies included in this review (Figure 1), the noise-reducing mechanism can be a Helmholtz resonator, Fabry–Pérot resonator, rainbow-trapping resonator, or Fano-like interference. These mechanisms are separately reviewed in that order from Section 3.1 to Section 3.4.

### 3.1. Helmholtz Resonator

The use of a micro-perforated cavity is a well-established approach to achieving broadband sound absorption [27]. Such cavities can be seen as being assembled from arrays of Helmholtz resonators that share the same cavity (or sometimes different ones). Therefore, the noise-reducing mechanism is based on the working principle of the Helmholtz resonator [28]. This approach is straightforward to implement and absorbs better at higher frequencies [29]. As it is a well-established approach, only one work on it was published in the last five years. Li et al. [30] documented a prototype involving two opposite-facing micro-perforated cavities in which one was placed inside the other. Although it was not measured, the prototype was believed to provide good ventilation by being 70% open in the frontal area. Experimentally, the best configuration could absorb 60–70% of the noise at 800–1000 Hz. As expected, good sound absorption occurs in the higher-frequency range. Although 60–70% of sound absorption may be regarded as acceptable performance, we will see in the remaining studies that the common reference value used to determine good performance is 80% or more.

Traditional Helmholtz resonators are reliable in providing good sound absorption in the lower-frequency range (<800 Hz)—albeit narrowband. The main limitation is that the resonator tends to occupy more space because of the large cavity size needed to lower the spring stiffness. Alternatively, a longer neck can be considered to increase the vibrating mass. To achieve good broadband sound absorption, the collective effect of multiple uniquely designed resonators is necessary. A straightforward approach is to create a metamuffler by arraying a series of resonators sequentially along the direction of noise propagation. Gao et al. [31] reported an example that involved 10 resonators with increasing cavity size. Every cuboid cavity was directly connected to the duct via a slit (Figure 2a). By using a genetic algorithm, the cavity sizes were optimized so that the different resonances would fall within 500–1000 Hz. In this frequency range, the measured sound absorption of the metamuffler was at least 80%.

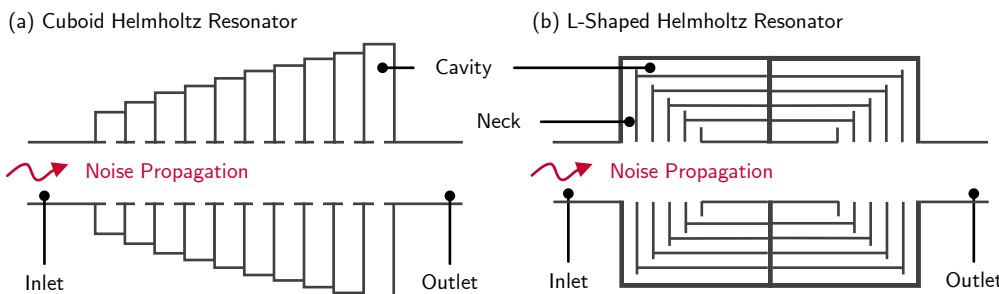

**Figure 2.** Schematic diagram (cross-sectional side view) showing the noise-reducing mechanisms in reported metamufflers—(**a**) a cuboid Helmholtz resonator [31] and (**b**) L-shaped Helmholtz resonator [32].

Other than changing the cavity size along the sequential array, Nguyen et al. [32] reported a prototype that varied the neck length as well. As a proof of concept, the design process started with the numerical study of one resonator. By considering an L-shaped resonator, the prototype was able to remain compact without any redundant interior space that had no influence on the overall performance of the metamuffler [33]. Through parametric studies, Nguyen et al. [32] noted five configurations that were eventually combined to form a two-layer metamuffler (i.e., 10 resonators) (Figure 2b). Experimentally, at least 40 dB of sound transmission loss was recorded at 650–850 Hz. Despite the remarkable performance, it is important to note that the performance may drop in a large-scale setup due to the influence of structural resonances. Guan et al. [34] and Ramos et al. [35] reported other prototypes that adopted the same design approach.

It is possible to design a metamuffler that performs well at low frequencies without involving complex structures. For example, Gao et al. [26] reported a resonator design that was realized with two separated layers. From the cross-sectional view (Figure 3a), the inner layer could be seen as an incomplete O-ring (i.e., C-shaped), while the outer layer could be seen as an O-ring. The air gap between both layers formed the cavity. The inner layer had an opening (normal to the direction of noise propagation) to permit sound wave interaction with the resonator. By sequentially arraying 10 unique resonators along the direction of noise propagation, the sound absorption was measured to be at least 80% at 380–470 Hz, which agreed well with the analytical solution.

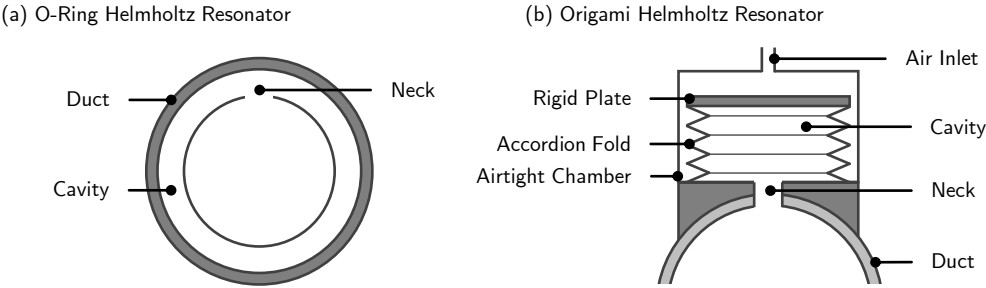

**Figure 3.** Schematic diagram (cross-sectional front view) showing the noise-reducing mechanisms in reported metamufflers—(**a**) an O-ring Helmholtz resonator [26] and (**b**) an origami Helmholtz resonator [36]. The direction of noise propagation is normal to the front view.

The advantage of complex designs is that more parameters are available for performance tuning. This feature is useful in practical applications because the same product can be implemented in different scenarios simply by tuning. One example was recently reported. Wen et al. [36] proposed an origami-based prototype in which the cavity of the resonator could be tuned by the air pressure to vary the volume. Four sides of the cuboid cavity were created by adopting an accordion fold in origami. The bottom was attached

to the duct with a small opening, allowing sound waves to enter the cavity. The top was covered with a rigid plate. An airtight chamber enclosed the resonator (Figure 3b). If air pressure was pumped into the chamber, the cavity would compress due to the accordion fold. By knowing the characteristics of the noise source, the air pressure could be accordingly varied to tune the resonator's narrowband performance within the working frequency range of 350–650 Hz. In this case, if the application required low-frequency noise control, a lower air pressure could be considered to enlarge the cavity, thus lowering the spring stiffness of the resonator.

Another means of performance tuning can be achieved by adopting modularity in the design of unit cells. For example, Xiang et al. [37] reported a prototype that could effectively absorb sound despite having a large open area of up to about 70%. The unit cell consisted of two parts—open and closed areas. The open area allowed unrestricted airflow, while the closed area provided the mechanism for sound absorption via modular cuboid blocks. Each block came with a slit at the incident surface that would guide the sound waves through a pair of weakly coupled split-tube resonators. At the opposite surface, the sound waves—now with a lower energy—would exit through another slit (Figure 4). To show how flexible the prototype was, three configurations involving seven blocks were investigated. Each block was carefully tuned to resonate at different frequencies so that the collective effect would lead to at least 80% sound absorption (measured) at 500–580 Hz.

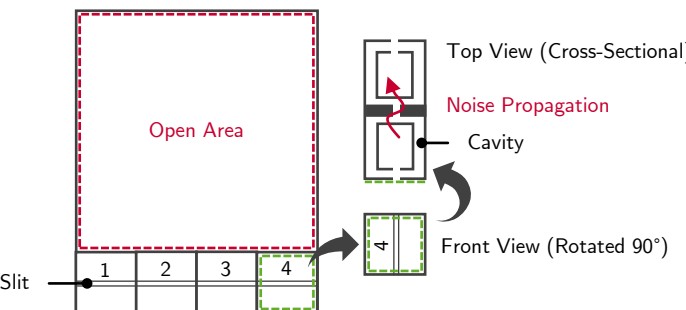

**Figure 4.** Schematic diagram showing the modular metamuffler proposed by Xiang et al. [37]. The direction of noise propagation is normal to the front view.

Instead of modular building blocks, the same research group proposed another way to achieve performance tuning. In the prototype [38], the cavity size could be manually compressed or expanded by pushing or pulling a sliding mechanism. This sliding process was analogous to using a syringe. With respect to the earlier prototype [37], the open area was kept similar at 69% (compared to 70%) in the present prototype. The largest and smallest cavity sizes were tested to achieve more than 80% sound absorption at 330–350 Hz and 490–520 Hz, respectively, agreeing well with the simulations.

### 3.2. Fabry–Pérot Resonator

Like in Helmholtz resonators, acoustic energy is dissipated due to the interaction of sound waves with the cavity. However, for low-frequency sound absorption to be effective, the final design often leads to one that comes with a large cavity and a long neck. This limitation can be overcome by Fabry–Pérot resonators. In this kind of resonator, the cavity is formed by folding or coiling narrow channels within the given design space. Owing to this construction, Fabry–Pérot resonators are sometimes referred to as "space-coiling resonators" [39] or "labyrinth resonators" [40]. As sound waves are forced to propagate over a long distance in the extended cavity, more acoustic energy can be dissipated [41].

Shao et al. [42] demonstrated effective low-frequency sound absorption with a metamuffler that boasted a large open area of about 50%. The metamuffler was made up of three resonators placed sequentially along the interior wall of the duct. In all resonators, the number of folds (two) along the channel was kept identical. Hence, the characteristics of every resonator were solely governed by the dimensions of the folded channel (Figure 5a).

In their work, Shao et al. [42] focused on varying the damping factor of the resonators—low, medium, and high. Under plane waves at normal incidence, impedance test results showed that the best configuration could absorb more than 80% of the noise at 300–340 Hz. With plane waves at oblique incidence, the performance was numerically shown to drop by about 10% in the same frequency range. Concurrently, Wang and Mao [43] published a study of a similarly designed metamuffler. The results of the work showed consistency with those of Shao et al. [42]—that is, more than 80% narrowband sound absorption at low frequencies (350–400 Hz). Consistent observations at 375–400 Hz were also made by Du et al. [44] based on a single-fold concept.

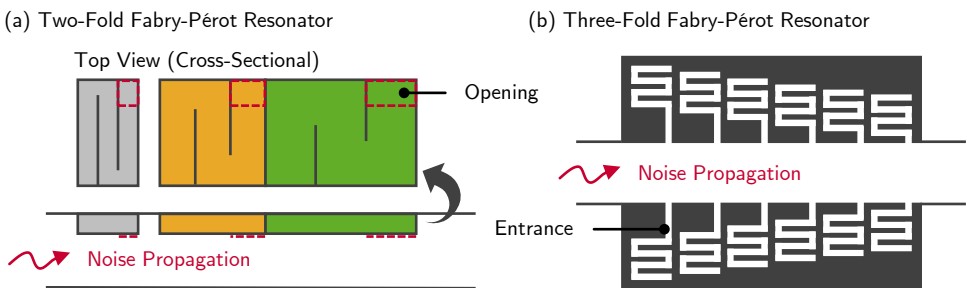

**Figure 5.** Schematic diagram showing the noise-reducing mechanisms in reported metamufflers—(**a**) a two-fold Fabry–Pérot resonator [42] and (**b**) a three-fold Fabry–Pérot resonator [45].

Like in Helmholtz resonators, it is possible to widen the frequency bandwidth with a good sound absorption by considering more unique resonators in the metamuffler [46,47]. For example, Yang et al. [46] consecutively stacked 10 unique resonators together to form a metamuffler with an open area of 25%. According to a parametric analysis, the final combination could absorb more than 80% of the noise at 600–850 Hz. Although the work successfully demonstrated bandwidth widening, the iterative design process could be laborious to follow in real-world application.

Speaking of real-world applications, Kim et al. [45] identified how a heating, ventilation, and air conditioning (HVAC) system could cause interior noise issues in electric vehicles. To innovatively address this issue, the research group collaborated with Hyundai Motor Group to study various metamuffler designs as potential solutions. The shape and size of the metamuffler were constrained by an actual HVAC system (rectangular duct) that was retrieved from a vehicle. The metamuffler had four attachments—one on each wall. The same array of six resonators was considered for each pair of opposite-facing attachments. As with what Shao et al. [42] considered, Kim et al. [45] kept the number of folds (three) identical in all resonators (Figure 5b). By varying the length of the folds and the entrance length in each pair of opposite-facing attachments, the measured insertion loss could reach up to 10 dB in the targeted frequency range (800–1000 Hz). When airflow was introduced into the HVAC system, the performance was found to drop, reaching up to only 5 dB in the same frequency range.

At this juncture, only the prototypes proposed in three different works [42–44] showed good sound absorption at a much lower frequency range in contrast to the other works [45–47]. Liu et al. [48] proposed another prototype that could reduce sound transmission by more than 10 dB at a frequency range that was even lower (190–430 Hz). In their work, the metamuffler was formed by sequentially stacking 10 resonators with unique dimensions. Each resonator was designed to resemble a circular tile with a through-hole at the center. Along the wall of the through-hole, the sound waves would enter through a small opening and interact with the cavity. The cavity made use of seven structural branches for space-coiling (Figure 6). The desired characteristics of the metamuffler were obtained by varying three parameters—the length of the branches, the cavity depth, and the opening location. If the resonator had its branches removed, it would then function like a Helmholtz resonator, as demonstrated by Kumar et al. [49].

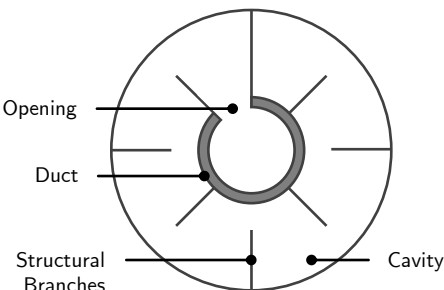

**Figure 6.** Schematic diagram (cross-sectional front view) showing the metamuffler proposed by Liu et al. [48]. The direction of noise propagation is normal to the front view.

In another similar design, Kumar and Lee [50] divided the cavity into six smaller ones. As such, the wall of the through-hole now had six openings—instead of one—positioned equidistantly. Behind every opening, sound waves would propagate along a four-fold channel, losing energy from multiple reflections. When tested in an impedance tube, the prototype could absorb at least 80% of the noise at 850–950 Hz. Kumar and Lee [51] recommended considering more folds for the absorption bandwidth to occur at lower frequencies.

### 3.3. Rainbow-Trapping Resonator

The term "rainbow trapping" is borrowed from optics, in which different wavelengths of electromagnetic waves are trapped in an array of gradient structures within a single device. A frequency-dependent interaction disperses the incident waves and slows down the wave velocity, leading to high broadband sound absorption [52]. The gradient structures must be sufficiently rigid so that no structural resonances can interact with the trapped waves, making the system less effective [53].

The consideration of rainbow-trapping resonators in metamufflers has been progressively studied by one research group [54–56]. At the start, a metamuffler was designed by varying the number of gradient cavities along the duct in descending order (i.e., from large to small along the direction of noise propagation). By comparing the simulated results, Fusaro et al. [54] selected the most ideal configuration for further studies. Subsequently, the metamuffler was treated as a unit cell and arrayed eight times in a continuous circular manner to form a tile-like sound barrier. A rigid panel was adhered to one side of the sound barrier, minimizing direct sound transmission into the receiver end. Instead, for the incident waves to reach the receiver end, they were forced to propagate through the side openings sandwiched between the gradient cavities (Figure 7a). Although sound transmission through the rigid panel is still possible, the work omitted this effect and assumed the rigid panel to be perfectly reflective. Numerically, the prototype could reduce sound transmission by more than 10 dB at 0.5–5 kHz, greatly surpassing the performance of the initial unit cell (more than 10 dB of sound transmission loss at 1.6–1.8 kHz). Expectedly, the boundary conditions were found to significantly affect the results. However, as this was a numerical work, the conclusions could be stronger if measurements were conducted. Soon, the research group [55] reported a lab study for a prototype with four side openings instead of eight. Hence, the measured results could not provide a fair comparison with those in the previous work [54]. Nonetheless, with four side openings, the best configuration could still demonstrate more than 10 dB of insertion loss at a slightly narrower frequency range (1.1–5 kHz). Later, the research group [56] evaluated the same prototype with respect to human perception.

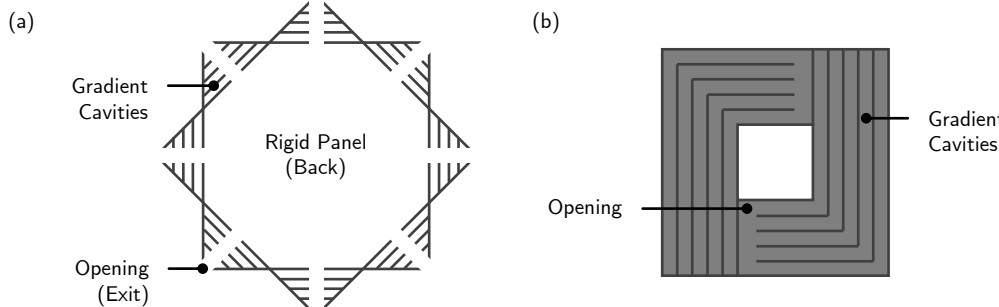

**Figure 7.** Schematic diagram (cross-sectional front view) showing the tile-like metamufflers proposed by (**a**) Fusaro et al. [54] and (**b**) Yang et al. [57]. The direction of noise propagation is normal to the front view.

Yang et al. [57] put forth another tile-like prototype created from two sets of L-shaped gradient structures. Both sets were oriented in an asymmetrical manner such that the center of the prototype had a square opening (Figure 7b). Through simulations, the wave velocity of sound waves with certain wavelengths was shown to drop (near-zero). With the right dimensions, the prototype could reduce sound transmission by more than 10 dB at 0.65–1.6 kHz, agreeing well with the simulation.

### 3.4. Fano-like Interference

Unlike in the resonator concepts presented earlier, the use of Fano-like interference as the noise-reducing mechanism in metamufflers can be challenging to design. For resonators, broadband noise reduction is generally achieved by stacking multiple unit cells with dissimilar dimensions. To realize Fano-like interference, the metamuffler must be designed with two paths—one to transmit the sound waves directly (i.e., unscattered) and one to scatter the sound waves via local resonance. At the outlet, the interactions between the scattered and the unscattered sound waves lead to constructive and destructive interference [58]. If the interference is destructive, noise is reduced. For this phenomenon to happen, the unit cell must be asymmetric. Because an unscattered path must be present, the use of Fano-like interference is highly viable in metamufflers, which are mostly designed with an opening for airflow.

One research group [59,60] showed that, albeit challenging, it is still possible for metamufflers to reduce sound transmission via Fano-like interference. The prototype had a through-hole at the center for direct transmission, propagating unscattered sound waves at the receiving end. Surrounding the exterior wall of the through-hole was a helical channel that would scatter the sound waves. By varying the pitch of each spiral, the prototype could realize Fano-like interference and eliminate constructive interference. Experimentally, the phenomenon could reduce sound transmission by more than 10 dB at 0.9–1.4 kHz, agreeing well with the simulation.

## 4. From Metamufflers to Metapanels

Not all metamuffler designs are suitable for scaling up. In the preceding section, the reviewed studies showed that metamufflers with tile-like unit cells can potentially be scaled up to form a sound-insulating panel. The scaling process involves arraying the unit cells in two directions. Let us look at the unit cell proposed by Yang et al. [57] as an example (Figure 7b) to elaborate further. With respect to the figure, imagine stacking the same unit cell repeatedly in the horizontal and vertical directions. The assembly will eventually resemble a panel. In the literature, many creative ways of scaling up without relying on permanent adhesive have been proposed. For instance, a puzzle joint can be designed along the edges of each unit cell [49]. Every unit cell can then be securely joined together. Alternatively, a continuous rigid frame with slots can be machined to host the unit cells [61]. The naming conventions of such assemblies vary in the literature. They are commonly

referred to as "metapanels" [61], "metabarriers" [60], or "metasurfaces" [62]. To place more emphasis on ventilation, the term "metawindow" was recently introduced [56]. Essentially, all of these terms are used to describe the same assembly. For consistency in writing, the term "metapanel" is used in this review.

As discussed in the preceding section, metamufflers are implemented by replacing a finite section of the ducting system. For metapanels, the general approach to implementing them is similar except that in this case, the application is for a wall rather than a ducting system. The major limitation is that to fabricate a metapanel, huge resources are required in terms of personnel and money [63]. Therefore, the number of studies included in this review is not large (Figure 1). Even so, we will see later that the concept of scaling up to form a metapanel has mainly been proposed numerically. Experimental work is still being conducted based on the metamuffler design. The noise-reducing mechanisms are separately reviewed in Section 4.1 to Section 4.3.

### 4.1. Helmholtz Resonator

Wu et al. [64] proposed a metapanel formed by arraying a pair of split-tube resonators in one direction. The pair of resonators had identical dimensions. One of them was oriented to face the incident wave, while the other was oriented to face the receiving end. The overall design was similar to that proposed by Xiang et al. [37] (Figure 4), except for the orientation of the resonators. Numerically, Wu et al. [64] found that it is important to consider multiple layers for higher sound absorption over a wider frequency range. For instance, by increasing the number of layers, the metapanel could absorb more than 80% of the noise at 320–370 Hz (three layers) instead of 340–360 Hz (one layer). To validate this, one pair of resonators (i.e., one unit cell) was tested in an impedance tube. To demonstrate the superiority, the sound absorption performance was compared to that of a commercial sound foam. In the comparison, the pair of resonators was found to outperform the commercial sound foam by around 60% at 300–380 Hz. Outside of this frequency range, the commercial sound foam was superior.

Among the reviewed studies, only Yu [62] performed an evaluation via in situ testing. Made up of $3 \times 3$ unit cells, the double-layer metapanel was 76 cm long, 55 cm wide, and 5 cm thick. These dimensions were necessary to replace one of the three panels of the casement window with the metapanel. The unit cells were simply designed according to a typical Helmholtz resonator (i.e., one opening and one cavity). To ensure a fair performance comparison between the metapanel and the original casement window, the latter was opened up to the point in which the open area (not provided) was the same as that of the former. Overall, the metapanel was shown to outperform the casement window by up to 10 dB in the tested frequency range (100–3150 Hz).

### 4.2. Bragg Diffraction

Specifically for high-frequency noise, Wang et al. [65] suggested considering Bragg diffraction as the noise-reducing mechanism. To achieve this phenomenon, the metapanel had to be formed by periodic structures that were separated at a distance governed by the wavelength of interest and the angle of scattering. Wang et al. [65] proposed building a metapanel by arraying square tubes with a cross-channel in one direction (Figure 8a). Following topological optimization of the periodic structures (Figure 8b), the simulated results showed that the metapanel could reduce sound transmission by more than 10 dB across a wider frequency range. Comparing the initial design and the optimized design, the frequency range with good performance widened by 850 Hz from 2.2–5.1 kHz to 1.5–5.25 kHz. This outcome was consistently observed in the measured results.

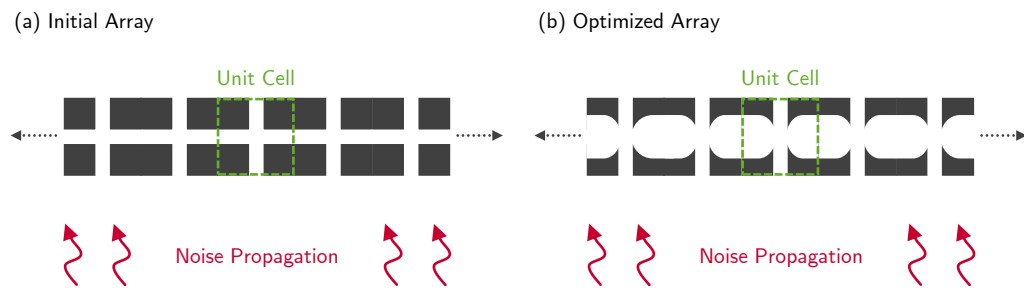

**Figure 8.** Schematic diagram (cross-sectional top view) showing the metapanel proposed by Wang et al. [65] (**a**) before and (**b**) after topological optimization.

### 4.3. Fano-like Interference

Nguyen et al. [66] proposed a three-layer metapanel that could reduce sound transmission by at least 10 dB at 0.45–3.85 kHz. Each increasing layer consisted of space-coiling structures with an increasing number of folds. Although the unit cell can essentially be classified as a Fabry–Pérot resonator, Nguyen et al. [66] attributed the overall performance to Fano-like interference, which is the interaction between scattered and unscattered sound waves at the transmitted side of the metapanel. Although only one unit cell of the three layers was tested in an impedance tube, good agreement with the simulation was observed. Additionally, Nguyen et al. [66] experimentally demonstrated the importance of treating the inner walls of the space-coiling structures with sound-absorbing materials to enhance high-frequency sound absorption. Alternatively, the unit cells could be specially designed with a sliding mechanism for the overall performance of the metapanel to be easily configurable according to the signature of the noise source [67].

## 5. From Metapanels to Metacages

Mirzaei et al. [68] first introduced the theoretical concept of metacages for biological applications. The metacage was realized with thin shielding structures made of long nanowires. Potentially, the structures could not only protect living microorganisms from electromagnetic radiation, but also sustain life by permitting the exchange of fluid and gas with the surroundings. Analytically, Mirzaei et al. [68] successfully demonstrated the shielding effect in two scenarios—preventing any electromagnetic radiation from either entering or leaving the metacage. Soon, more researchers became interested in the phenomenon and started to advance the concept of the metacage for optical applications [69]. However, studies on metacages for similar applications have become limited in the last five years. As raised by Qian et al. [69], the decline in the reported studies might be due to how difficult it is to fabricate specimens for experiments.

In addition to shielding from electromagnetic waves, metacages have also been studied for shielding from sound waves. In this context, they are widely referred to as "acoustic metacages". Unlike metacages designed for optical applications (nanoscale), acoustic metacages do not face the same manufacturing challenges because they are on the macroscale. In other words, acoustic metacages can be easily fabricated via traditional and additive manufacturing techniques. For complex designs, they can be assembled from smaller components produced with different manufacturing techniques.

Acoustic metacages are essentially acoustic enclosures. Both solutions primarily serve to prevent noise from either entering or leaving the encompassing space. The former (latter) applies when the noise source is outside (inside) the acoustic metacage or enclosure. Although the solutions can work either way, it is more common for the noise source to be within the encompassing space [70]. Examples include machines and engines. What is the main feature that differentiates both solutions? Acoustic enclosures (Figure 9a) are designed to surround the noise source with walls that have no gaps, thus preventing any fluid exchange with the exterior space. The walls possess strong insulating properties that reduce

the propagation of noise into the surroundings. Acoustic metacages (Figure 9b) are meant to augment acoustic enclosures by permitting fluid exchange with the exterior space while still providing adequate noise reduction. Fluid exchange is made possible by openings that directly connect both interior and exterior spaces. Noise reduction is achieved with specially designed structures that are repeatedly placed around the encompassing space.

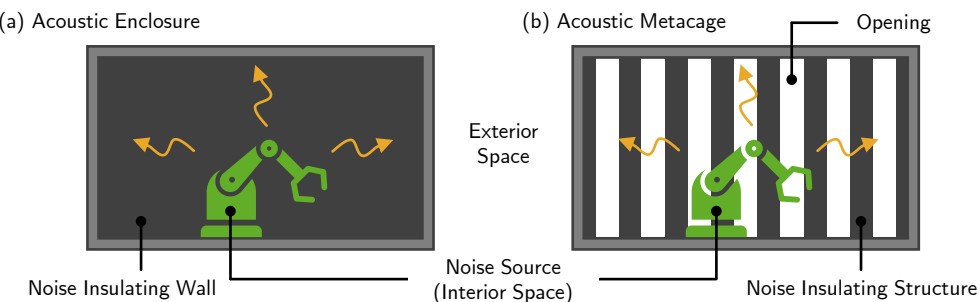

**Figure 9.** Schematic diagram (cross-sectional front view) showing a typical (**a**) acoustic enclosure and (**b**) an acoustic metacage encompassing a noise-emitting machine.

As the concept of metacages for optical applications was first put forth in 2015 [68], studies on acoustic metacages have been reported only since 2018. Shen et al. [71] first reported an acoustic metacage with radially arrayed unit cells consisting of an open channel and four shunted Helmholtz resonators (Figure 10a). Special design considerations were given to the resonators such that the phase difference between each succeeding resonator was $\pi/2$ rad. At the end of the open channel, the phase of the transmitted sound waves would be shifted by $2\pi$ rad relative to that of the impinging sound waves. At the targeted frequency of 2.5 kHz, the three-dimensionally printed prototype achieved an average sound transmission loss of about 10 dB. Numerically, the performance was overestimated by 4 dB (i.e., 14 dB). In terms of ventilation, the prototype permitted 38% of the airflow to exit.

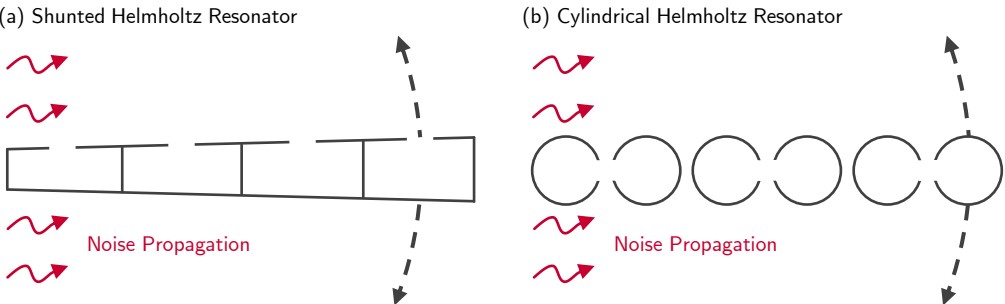

**Figure 10.** Schematic diagram (cross-sectional top view) showing the unit cell designs in reported metacages—(**a**) a shunted Helmholtz resonator [71] and (**b**) a cylindrical Helmholtz resonator [72].

Instead of shunted Helmholtz resonators, Krasikova et al. [72] considered three pairs of cylindrical Helmholtz resonators as one unit cell (Figure 10b). The three-dimensionally printed resonators were radially and equidistantly arrayed to form a toroid, allowing at least 10% of the airflow to exit. Each resonator was represented by one hollow tube with a slit extending along the longitudinal axis. Parametric studies showed that the most desirable outcome was achieved when each pair of resonators had their slits directly facing each other. The prototype was evaluated based on the average sound transmission loss (19 dB) over a broad frequency range (1.5–16 kHz), rather than a targeted frequency (not provided), which is common for Helmholtz resonators.

Melnikov et al. [73] first proposed the practical use of a metacage to reduce stage machinery noise specifically at 613 Hz. At the preliminary stage, the proof-of-concept design was three-dimensionally printed and experimentally shown to reduce sound trans-

mission at 1.5 kHz (targeted frequency) by about 14 dB—1 dB more than the simulation (15 dB). The cuboid assembly had three faces (two adjacent sides and top) with unit cells arrayed equidistantly and two adjacent side faces applied with sound absorption materials. The bottom face was treated as rigid. Each unit cell consisted of one cylindrical Helmholtz resonator—similar to those reported by Krasikova et al. [72]—with its slit oriented to face the encompassing space. Motivated by the promising outcome, Melnikov et al. [73] reconfigured the preliminary design such that the size could encompass the stage machine, and the targeted frequency became 613 Hz instead of 1.5 kHz. In the new assembly, only one of the faces was occupied by the unit cells. The remaining five faces were applied with sound absorption materials to provide noise reduction at high frequencies. At 613 Hz, the sound transmission loss was about 14 dB—10 dB higher than that in the simulation (24 dB). Numerically, the presence of ventilation was shown only by comparing the amount of heat dissipated by the prototype and an acoustic enclosure.

Kumar and Lee [51] increased the design complexity by three-dimensionally printing additional structural features within every cylindrical Helmholtz resonator. In the first concept, every resonator was equally divided into two halves via an internal partition. By adding a slit on both halves, the unit cell was then represented by a two-sided half-cylindrical Helmholtz resonator. The second concept was built upon the first concept by extending two branches (60° apart) from the center of the cylinder up to a certain point along the radial direction such that the half-cylindrical cavity was not further divided into smaller volumes. Regardless of the design concept, higher noise reduction was measured when the slits were directly facing each other compared to when they were facing the encompassing space. Kumar and Lee [51] claimed that the higher noise reduction could be attributed to more interaction between the two cavities and the exiting sound waves. Otherwise, the interaction would mainly involve the source-facing cavity. At the targeted frequency of 2.5 kHz, the second concept performed the best with an insertion loss of 12 dB. The first concept performed more poorly by 2 dB (10 dB).

Concurrently, Kumar et al. [74] proposed another metacage without relying on additive manufacturing. The cylindrical Helmholtz resonators were replaced by off-the-shelf aluminum profiles. As the profile dimensions were unsuitable for acting as Helmholtz resonators, the key mechanism that allowed noise reduction was Bragg diffraction. Although the prototype was cheaper to fabricate compared to that which was proposed separately [51], the lack of customization led to poor insertion loss (3 dB) at the targeted frequency (2 kHz).

There are many ways to introduce more design parameters to typical Helmholtz resonators. As discussed above [51], adding internal partitions or branches can be one of the ways to tune the cavity shape and size. Another way is to extend the passageway that leads to the cavity. As reported by Guan et al. [75], a coiled passageway could be considered so that sound waves would propagate for a longer distance before arriving at the cavity. A coiled passageway was formed by extending the lengthwise slit along the peripheral of a square hollow section. Three pairs of two square hollow sections represented one unit cell. Based solely on numerical results, Guan et al. [75] reported how the metacage could achieve less than 20% transmittance at the targeted frequency of 120 Hz. However, it is important to note that the work not only did not provide geometrical details of the metacage, it also did not conduct measurements to validate the simulated outcome.

Through the careful design of the unit cells, Chen et al. [76] reported the possibility of applying the working principle of Fano-like interference as the key mechanism for achieving noise shielding. Fano-like interference is the outcome of the interaction between scattered and unscattered sound waves along the propagation path. When a unit cell undergoes resonance, it scatters part of the impinging sound waves, causing interactions with the unscattered sound waves that propagate through the intermediate opening of adjacent unit cells. For this phenomenon to happen, the unit cell must be asymmetric. Complex structural features are not needed. A simple zigzag profile is sufficient (Figure 11a).

For instance, at the targeted frequency of 5.02 kHz, Chen et al. [76] measured an average sound transmission loss that was 15 dB—6 dB lower than that in the simulation (21 dB).

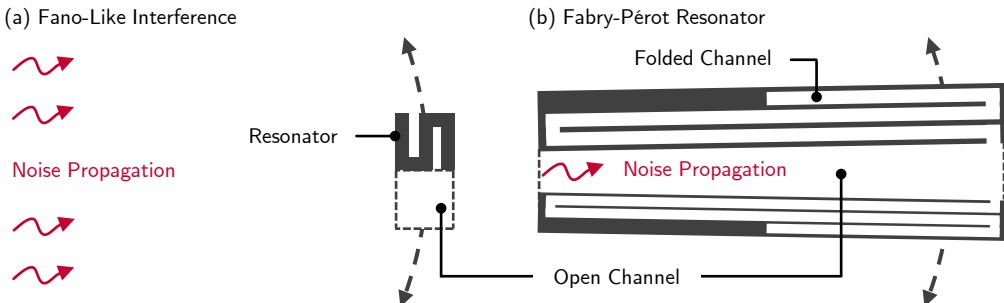

**Figure 11.** Schematic diagram (cross-sectional top view) showing the unit cell designs in reported metacages—(**a**) Fano-like interference [76] and (**b**) a Fabry–Pérot resonator [77].

Without relying on Fano-like interference, Yin et al. [78] demonstrated how acoustic rainbow-trapping unit cells could also be an option in the design of metacages. In their work, a unit cell was formed by a radial array of grooves sculpted with gradient branches. Due to the complex and small features (5.5 mm), the unit cells could only be fabricated via three-dimensional printing. The prototype was numerically and experimentally shown to provide more than 10 dB in average sound transmission loss at the targeted frequency range of 1.5–1.8 kHz. This performance was not limited to typical peripheral shapes (torus or toroid) of the metacage. Yin et al. [78] further showed that with rainbow-trapping unit cells, consistent performance could also be observed for a heart-shaped arrangement.

Recently, Long et al. [77] conceptualized a metacage formed by a radial array of two Fabry–Pérot resonators in one unit cell (Figure 11b). The intermediate opening between the two resonators permitted airflow to exit (not measured). Based on the objective of shielding noise at 260 Hz, the work employed a genetic algorithm to design the dimensions of both resonators. In a simulation, the prototype achieved at least 10 dB in average sound transmission loss. Although the concept was not experimentally studied as a whole, the three-dimensionally printed unit cell was found to perform consistently with the simulation.

So far, the measured performance has been determined based on a two-dimensional setup with an open top. Even if the top was covered, it was covered with a homogeneous panel [51,74]. Liu et al. [79] first attempted to cover the top and bottom of a metacage with the same unit cells arrayed along the peripheral. Each unit cell had an opening at the center for the airflow to exit. Along the wall of the opening were four smaller openings that served to direct sound waves into the labyrinthine channels of the Fabry–Pérot resonator. The prototype was not evaluated based on conventional metrics, such as insertion loss or sound transmission loss. Instead, it was evaluated based on the normalized sound intensity. The simulated and measured results showed that the prototype could perform consistently with and without the presence of low-speed wind (2 m/s).

## 6. Outlook

From the studies included in this review, we have seen how scaling up can allow metamufflers to be converted into metapanels, and then into metacages. Essentially, the successful realization of metacages relies on the fundamental work conducted on the unit cells in metamufflers. Due to the growing emphasis on adopting natural ventilation to reduce energy consumption in buildings, acoustic metamaterials with ventilation capabilities have started to emerge, showing favorable performance and promising applications. In particular, large-scale metacages can be useful in shielding noise, be it exterior or interior. In the former case, the receiver within the metacage can enjoy a quiet environment without exterior noise. In the latter case, the noise source within the metacage can be confined to minimize the propagation of noise into the surroundings. However, the route to the endpoint involves many technical challenges to be addressed before metacages can be truly

considered for real-world applications. The challenges can be identified from the summary of the key information for metacages reported over the last five years (2018–2023) (Table 1).

**Key mechanism.** Five of the reviewed studies (50%) considered a Helmholtz resonator as the noise-reducing mechanism. This popularity might be attributed to how the working principle can be easily understood compared to other noise-reducing mechanisms. Fundamentally, there are only two design parameters—namely, the neck and the cavity. By adjusting the dimensions, the resonant behavior can be tailored according to the characteristics of the noise source. This offers great design flexibility, through which the resonator can be considered in a noisy environment that is made up of mainly low-frequency noise or high-frequency noise. In the former case, the resonator should be designed with a wider neck and a larger cavity. Naturally, the resulting design comes with a high spatial penalty—a limitation that is generally not welcomed in real-world applications. To address this limitation, researchers are challenged to innovate and explore ventilated resonator designs that are not only compact, but also effective in low-frequency noise control. The literature has consistently established that Helmholtz resonators can be highly effective in low-frequency noise control. Even now, studies proposing innovative designs of Helmholtz resonators are still being reported [80,81]. This constant interest suggests that eventually, Helmholtz resonators may be more viable in real-world applications.

**Fabrication method.** Additive manufacturing is credited with allowing acoustic metamaterials—not only ventilated ones—to be fabricated for experimental studies. In Table 1, additive manufacturing clearly stands out as the preferred fabrication method among the reviewed studies. Omitting the two studies [75,77] that did not involve any experimental work, 87.5% of the reviewed studies fabricated unit cells via additive manufacturing. Only one study (12.5%) fabricated unit cells through traditional machining. Due to the limited printing size, every unit cell can only be created individually. Subsequently, the unit cells are manually pieced together to form the metacage. This limitation becomes a challenge during the process of scaling up. If a larger number of unit cells is required to form a metacage, the process will be laborious and time-consuming. It is possible to address this challenge in the future when collaborative printing becomes affordable and accessible [82].

**Prototype size.** The largest prototype size was reported by Melnikov et al. [73] because the purpose was targeted at stage machinery in order to reduce the emission of noise into the surroundings. The work reported promising results, showing how metacages are suitable for real-world applications. Naturally, the larger a metacage is, the wider the range of applications will be. However, unless larger metacages are built and studied, it is now unclear whether the noise-shielding performance will be maintained as the metacages expand in size. This challenge is tied to the present limitation of additive manufacturing, as discussed in the preceding paragraph. Otherwise, as shown by Kumar et al. [74], machined profiles will have to be considered.

**Performance.** The performance of metacages has been determined in terms of sound transmission loss or insertion loss. As the purpose of metacages is for noise shielding, it is expected that none of the reviewed studies evaluated the metacages in terms of the sound absorption coefficient. To date, low-frequency noise remains challenging to mitigate. In Table 1, Helmholtz resonators and Fabry–Pérot resonators were shown to be viable options to consider in metacages meant for low-frequency noise control. As discussed in the preceding paragraph, there is a need to study metacages at a large scale to determine if good performance can be maintained, or at least without any significant drop. In future studies, researchers are recommended to quantitatively determine the ventilation performance. Instead of simply reporting the open area, this information can help justify how effective a metacage is in providing ventilation. Among the reviewed studies, only three of them (30%) had done so.

**Table 1.** Summary of the key information for metacages reported over the last five years (2018–2023). In the last three columns, a tick indicates that the corresponding work was done strictly on a metacage (not just on one unit cell).

| Source | Key Mechanism | Fabrication Method | Prototype Size [mm] [1] | Targeted Frequency | Peak Performance (Measured) [2] | Peak Airflow (Measured) | Simulation | Lab Test | In Situ Test |
|---|---|---|---|---|---|---|---|---|---|
| [71] | Helmholtz Resonator | Additive | R150, H45 | 2.5 kHz | ≈10 dB (STL) | 38% | ✓ | ✓ | ✗ |
| [72] | Helmholtz Resonator | Additive | R300, H60 | 1.5–16 kHz | 19 dB (STL) | ≥10% | ✗ | ✓ | ✗ |
| [73] | Helmholtz Resonator | Additive | L600, W590, H530 | 613 Hz | 14 dB (STL) | Not Measured | ✓ | ✓ | ✓ |
| [51] | Helmholtz Resonator | Additive | R250, H154 | 2.5 kHz | 12 dB (IL) | Not Measured | ✗ | ✓ | ✗ |
| [74] | Bragg Diffraction | Traditional | L360, W360, H220 | 2 kHz | 3 dB (IL) | Not Measured | ✗ | ✓ | ✗ |
| [75] | Helmholtz Resonator | Not Fabricated | Not Fabricated | 120 Hz | Not Measured | Not Measured | ✓ | ✗ | ✗ |
| [76] | Fano-Like Interference | Additive | Not Fabricated | 5.02 kHz | 15 dB (STL) | Not Measured | ✓ | ✓ | ✗ |
| [78] | Rainbow Trapping Resonator | Additive | R450, H50 | 1.5–1.8 kHz | ≥10 dB (STL) | 58% | ✓ | ✓ | ✗ |
| [77] | Fabry-Pérot Resonator | Not Fabricated | Not Fabricated | 260 Hz | Not Measured | Not Measured | ✓ | ✗ | ✗ |
| [79] | Fabry-Pérot Resonator S | Additive | L164, W164, H164 | 1.03 kHz | Not Measured | Not Measured | ✓ | ✓ | ✗ |

[1] R, L, W, and H denote the radius, length, width, and height, respectively. [2] STL and IL denote the sound transmission loss and insertion loss, respectively.

**Mode of study.** Presently, metacages have mostly been tested in laboratories. As a laboratory is usually a controlled environment, the performance of the metacage might differ from in situ test results. Among the reviewed studies, only one work [73] considered in situ testing. The remaining nine studies (90%) only considered laboratory testing. Although the proof of concept—as with metamufflers—can be shown based on one unit cell, future studies are recommended to consider in situ testing for the research community to determine whether metacages are indeed viable in real-world applications.

## 7. Conclusions

To conclude, the growing emphasis on sustainability as one of the main pillars in developing future cities motivated this timely systematic review of emerging ventilated acoustic metamaterials for noise control. By limiting the scope to cover only the last five years (2018–2023), we classified ventilated acoustic metamaterials into three kinds—metamuffler, metapanel, and metacage. Generally, investigations started from a metamuffler. If favorable results were measured, the metamuffler could be scaled up to a metapanel. Eventually, the metapanel could be further shaped into a metacage. Although the concept of metacages was only recently reported, promising noise-shielding capabilities have been demonstrated. If more research efforts are invested, we may see real-world applications of metacages as a novel ventilated noise control measure in the future.

**Author Contributions:** Conceptualization, L.Y.L.A. and H.P.L.; Data curation, L.Y.L.A.; Formal analysis, L.Y.L.A.; Funding acquisition, K.-M.L. and H.P.L.; Methodology, L.Y.L.A. and H.P.L.; Project administration, K.-M.L. and H.P.L.; Resources, F.C. and H.P.L.; Supervision, F.C. and H.P.L.; Visualization, L.Y.L.A.; Writing—original draft, L.Y.L.A.; Writing—review and editing, L.Y.L.A., F.C., K.-M.L., and H.P.L. All authors have read and agreed to the published version of the manuscript.

**Funding:** K.-M. Lim and H.P. Lee received financial support from the Ministry of Education, Singapore, under its Academic Research Fund Tier 2 (MOE-T2EP50221-0018). L.Y.L. Ang and F. Cui did not receive any specific grants from funding agencies in the public, commercial, or not-for-profit sectors.

**Institutional Review Board Statement:** Not applicable.

**Informed Consent Statement:** Not applicable.

**Data Availability Statement:** Not applicable.

**Acknowledgments:** We appreciate the digital library access provided by A*STAR and NUS.

**Conflicts of Interest:** We declare that there are no known competing financial interest or personal relationships that could have appeared to influence the work reported in this paper.

## Abbreviations

The following abbreviations are used in this manuscript:

| | |
|---|---|
| HVAC | Heating, Ventilation, and Air Conditioning |
| IL | Insertion Loss |
| IPCC | Intergovernmental Panel on Climate Change |
| STL | Sound Transmission Loss |

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
