# Peer review of "A Systematic Review of Emerging Ventilated Acoustic Metamaterials for Noise Control"

_sustainability, doi:10.3390/su15054113_

Round 1

Reviewer 1 Report

Dear authors,  the organization in the manuscript is correct and clear, starting from a description of the type – main focus, main objective - of the review that was to be done, to how references were selected and, finally, why they were considered or not. The organisation of manuscript around the three main categories encountered in bibliography seems adequate as it gives potential readers a clear, orderly and summarised vision - overview - of current state of research and innovation in the field.

Other authors follow a different pattern in similar works, for example that of Nansha Gao et al. (https://onlinelibrary.wiley.com/doi/full/10.1002/admt.202100698) - not included in your references, by the way - which distinguishes between active and passive noise control. It may not be a work so specifically focused on the issue of “ventilation”, but it does include in its analysis works oriented to that topic. Why has it not been included in your review? (“Advanced Materials Technologies” seems to be a good quality journal from Wiley - impact factor of 8.856)

Anyway, I consider your approach is correct and clear and the overall output of your revision of great interest and usefulness for other researchers, both for the analysis of the work done in recent years and for the search of possible synergies between researchers. It's been a long time since I've reviewed such a good article, both in content and form. Congratulations, in my opinion and to my best knowledge, you have done a great job.

Author Response

Please kindly refer to the attached response letter. Thank you.

Reviewer 2 Report

The review paper entitled “A systematic review of the emerging ventilated acoustic metamaterial for noise control” contains useful and fluently written information recovered from selected papers.

The building up of the paper is correct, but I strongly suggest shifting Figure 1 and chapters 2.1-2-3 into the supplementary material, because the description of the paper selecting strategy does not supply any important professional information about the topic.  

From the other points of view, the paper is written well, the building up of that is clear, and the reviewed pieces of information are useful.

Only a remark, that the neglection of so-called “non-reviewed” papers is not always useful, because the  “non-reviewed” paper’s information does not mean decisively a bad one. or incorrectly or not professionally written papers. I suggest in the future selecting the papers after evaluation of their content and deciding whether those are reviewed or not. But, that is decisively the right of the author to select the papers to involve in their review.

Author Response

(The authors gave the same response as above.)

Reviewer 3 Report

The authors systematically review the existing efforts towards acoustic metamaterials for noise control applications with ventilation within the past five years. The included topics are mainly focused on the working mechanisms of the existing metamuffler, metapanel, and metacage and their potential applications. Providing efficient noise cancelation with maintaining ventilation is of great significance in terms of reducing energy consumption, providing indoor acoustic comfort, and even aesthetics. Overall, the review content is meaningful and could have impact in the field of noise control. the manuscript is also well organized. I do not find any noticeable mistakes or misleading. Given all these considerations, I think the manuscript should be qualified for publication in Sustainability.

Author Response

(The authors gave the same response as above.)
